# Age Estimation Using Maxillary Central Incisor Analysis on Cone Beam Computed Tomography Human Images

**DOI:** 10.3390/ijerph192013370

**Published:** 2022-10-16

**Authors:** María Arminda Santos, Juan Muinelo-Lorenzo, Ana Fernández-Alonso, Angelines Cruz-Landeira, Carlos Aroso, María Mercedes Suárez-Cunqueiro

**Affiliations:** 1Department of Dental Science, University Institute of Health Sciences (IUCS), Cooperativa de Ensino Superior Politécnico e Universitário (CESPU), 4585-116 Gandra, Portugal; 2Department of Surgery and Medical-Surgical Specialties, Medicine and Dentistry School, Universidade de Santiago de Compostela, 15782 Santiago de Compostela, Spain; 3Department of Forensic Sciences, Pathology, Ginecology and Obstetrics, and Pediatrics, Medicine and Dentistry School, Universidade de Santiago de Compostela, 15782 Santiago de Compostela, Spain; 4Oral Pathology and Rehabilitation Research Unit (UNIPRO), University Institute of Health Sciences (IUCS), Cooperativa de Ensino Superior Politénico e Universitario (CESPU), 4585-116 Gandra, Portugal; 5Health Research Institute of Santiago de Compostela (IDIS), Complexo Hospitalario Universitario de Santiago de Compostela (CHUS/SERGAS), 15706 Santiago de Compostela, Spain

**Keywords:** dentistry, forensic, age estimation, cone beam computed tomography, CBCT

## Abstract

Forensic dentistry plays an important role in human identification, and dental age estimation is an important part of the process. Secondary dentin deposition throughout an individual’s lifetime and consequent modification in teeth anatomy is an important parameter for age estimation procedures. The aim of the present study was to develop regression equations to determine age in adults by means of linear measurements and ratios on sagittal, coronal and axial slices of maxillary central incisors using cone bean computed tomography (CBCT). Multiplanar measurements of upper central incisors were taken for a sample of 373 CBCTs. Subsequently, one-way analysis of variance (ANOVA) and multivariate linear regressions were performed for age estimation. The equations obtained from axial linear measurements and ratios presented a standard error of the estimate (SEE) of ±10.9 years (R^2^ = 0.49), and a SEE of ±10.8 years (R^2^ = 0.50), respectively. The equation obtained for multiplanar linear measurements presented a SEE of ±10.9 years (R^2^ = 0.52), while the equation for multiplanar ratios presented a SEE of ±10.7 years (R^2^ = 0.51). Thus, CBCT measurements on upper central incisors were found to be an acceptable method for age estimation. Horizontal measurements, especially pulp measurements, improve the accuracy of age estimate equations.

## 1. Introduction

Age estimation is an important aspect of forensic investigations and legal issues. Forensic techniques based on bone structures cease to be useful in the face of serious structural damage. However, seeing as they are very difficult to destroy, teeth are one of the main forensic tools for age estimation. Moreover, the literature suggests that dental indicators provide high accuracy for age estimation [1,2]. For these reasons, forensic dentistry is playing an increasingly relevant role. Laccasagne was the first to attempt age determination using dentition in 1889, and in 1925, Bodecker pointed out that certain morphological changes in teeth could be related to increased age [3,4]. Since then, a number of tooth characteristics have been used to determine age: (i) tooth wear, (ii) root dentin transparency, (iii) secondary dentin deposition, (iv) cementum annuli measurement, (v) Retzius striae count, (vi) aspartic acid racemization of dentin, (vii) radio carbon analysis, (viii) mineralization of third molars, and (ix)—with the development of 3D imaging techniques—pulp volume [5,6,7,8,9,10,11,12]. Methods based on biochemical and histological techniques which require tooth extraction are difficult to apply, time consuming and very expensive when there are many specimens to identify. Dental imaging techniques such as cone beam computed tomography (CBCT) can overcome many of these drawbacks. In addition, the fact that they enable long-term record storage and easy portability make these imaging techniques an unavoidable, valuable adjunct to forensic investigations [13].

Secondary dentin deposition is an important parameter for age estimation because it continues throughout life [14], and it can be indirectly measured with non-destructive imaging techniques [2,15,16]. Over one’s lifetime, secondary dentin deposition causes a decrease in the size of the pulp cavity. Based on this fact, various authors have estimated chronological age with imaging techniques [2,15,17]. Two-dimensional imaging systems are insufficient because they produce dimensional variations, are limited to mesiodistal dimensions, and analysis is hindered by factors such as malpositions and superpositions [13]. Thus, 3D imaging is essential for accurate forensic identification. Vanderwoort et al. were the first to apply 3D measurement methods to estimate age based on secondary dentin deposition [18]. Subsequent research analyzing linear measurements [19], areas [19,20,21] and volumes [11,18,22,23,24,25,26,27,28,29,30] have been performed using either micro-CT [22,31,32], CT [26,27], or CBCT [11,19,20,21,23,24,25,29,30]. Unlike previous research on the topic, the present study has included a much more thorough analysis involving multiple linear measurements and ratios in all three CBCT slices. We hypothesized that CBCT analysis of secondary dentin deposition by means of multiplanar linear measurements could yield convenient and useful methods for age estimation. The aim was to obtain regression equations for estimating age in adults through linear measurements and ratios on sagittal, coronal and axial slices of maxillary central incisors using CBCT.

## 2. Materials and Methods

### 2.1. Sample

A total of 373 randomly selected CBCTs were included in this cross-sectional study. The sample size was determined by means of the Slovin’s formula. These CBCTs were performed from 2012 to 2017 for treatment planning of various oral surgical procedures in the Radiology Unit of the Medicine and Dentistry School at the Santiago de Compostela University.

### 2.2. Inclusion and Exclusion Criteria

The inclusion criteria were the following: (1) patients older than 18 years, (2) maxillary with at least one upper central incisor, and (3) CBCT with a 0.3 mm minimum voxel size image quality. The exclusion criteria were as follows: (1) CBCT images which were impossible to rotate, blurred or incomplete, (2) decay, restoration or endodontic treatment, (3) ceramic/metal crowns or orthodontic elements, and (4) calcified root canal or root resorption.

### 2.3. Ethics Statement

Written informed consent was obtained from all patients. In compliance with ethical, methodological and legal requirements, this study was approved by the Galician Ethics Committee of Clinical Research (Ref: 2012/272). The methods were performed in accordance with relevant guidelines and regulations.

### 2.4. Image Evaluation

CBCT was carried out using i-CAT^®^ Model 17–19 (Imaging Sciences International Inc., Hatfield, PA, USA) with a flat-panel detector of amorphous silicon, an exposure protocol of 120 kVp and a current of 5 mA for 14.7 s. Patient occlusal plane was set parallel to the floor base by means of ear rods and a chin rest. DICOM files were reconstructed on computer (Samsung R522, Samsung Electronics, Seoul, Korea) using the 3D visualization software iCAT Vision^®^ v. 1.9. CBCT slice thickness was set at 0.25 mm.

### 2.5. Standardized Incisor Measurement Protocol

Multiplanar measurements (sagittal, coronal and axial) were performed by researcher 1 as follows:

#### 2.5.1. Sagittal Slice

Sagittal measurements at cementoenamel junction level. Linear measurements were obtained in the “incisor neutral position” which was achieved by placing the horizontal red line of the CBCT program at the level of cementoenamel junction of the upper central incisor (Figure 1a): (a) tooth width: from the most palatine point to the most vestibular point of the incisor (sce_TW, Figure 1b); (b) pulp width: from the most palatine point to the most vestibular point of the pulp (sce_PW, Figure 1b).Sagittal measurement at pulp horn level. In order to locate this level, the red horizontal line was moved from the cementoenamel junction to the pulp horn. After the pulp horn level was located, the incisor crown width was measured (sph_CW, Figure 1c).Sagittal vertical measurements: (a) root length: from the apex to the cementoenamel junction fooling the curvature of the tooth (s_RL, Figure 1d); (b) pulp chamber length: from the cementoenamel junction to the end of the pulp at crown level (s_PCL) (c) incisal length: from the incisal edge to the beginning of the pulp chamber (s_IL); (d) total tooth length: the sum of (a), (b) and (c) (s_TTL, sum of values 4 to 8, Figure 1d).

#### 2.5.2. Coronal Slice

Coronal measurements at cementoenamel junction level: (a) mesiodistal tooth length: from the most mesial point to the most distal point of the incisor (cce_MDTL, sum of the values 1+2+3); (b) mesiodistal pulp length: from the most mesial point to the most distal point of the pulp (cce_MDPL, Figure 2a).Coronal measurements at pulp horn level: (a) mesiodistal tooth length: from the most mesial point to the most distal point of the incisor (cph_MDTL, sum of the values 1+2+3, Figure 2c); (b) mesiodistal pulp length: from the most mesial point to the most distal point of the pulp (cph_MDPL, Figure 2c).

#### 2.5.3. Axial Slice

Axial measurements at cementoenamel junction level: (a) mesiodistal tooth length: from the most mesial and central point to the most distal and central point (ace_MDTL, sum of the values 1+2+3, Figure 2b); (b) mesiodistal pulp length: from the most mesial point to the most distal point of the pulp (ace_MDPL, Figure 2b); (c) palatovestibular tooth length: from the most vestibular-central point of the tooth to the most palatine-central point of the incisor (ace_PVTL, sum of the values 4+5+6, Figure 2b); (d) palatovestibular pulp length: from the most vestibular point to the most palatine point of the pulp (ace_PVPL, Figure 2b).Axial measurements at pulp horn level: (a) mesiodistal tooth length: from the most mesial point to the most distal point of the incisor (aph_MDTL sum of the values 1+2+3, Figure 2d); (b) mesiodistal pulp length: from the most mesial point to the most distal point of the pulp (aph_MDPL, Figure 2d); (c) palatovestibular tooth length; from the most palatine point to the most vestibular point of the incisor (aph_PVTL, Figure 2d).

Finally, for further analysis, ratios were calculated from the linear measurements obtained on the different slices and levels.

### 2.6. Intraobserver and Interobserver Variability

To check intraobserver variability, twenty randomly selected CBCTs were remeasured one month later by researcher 1. To check interobserver variability, researcher 3 measured the same twenty CBCTs and the results were compared with measurements made by researcher 1.

### 2.7. Statistical Analysis

Statistical analysis was performed using SPSS^®^v. 26.0 for Windows (IBM Corporation, Armond, NY, USA). Intraobserver and interobserver variability were assessed using intraclass correlation coefficient. Descriptive statistics were performed. The Kolmogorov–Smirnov test and the Levene test were applied to check for normality and the homogeneity distribution of the sample. One-way analysis of variance (ANOVA) with post hoc Bonferroni multiple comparison tests were used to compare linear measurements and ratios between age groups. Multivariate linear regression analyses were performed for age estimation based on sagittal, coronal and axial linear measurements and ratios. Statistical significance was set at *p* ≤ 0.05.

## 3. Results

From the initial 373 CBCTs, a total of 360 satisfied the inclusion criteria. Regarding the exclusion criteria, twenty-eight CBCTs were excluded because images were impossible to rotate, blurred or incomplete; twelve CBCTs presented incisors with decay, restoration or endodontic treatment; ten CBCTs showed crowns or orthodontic treatments; and four presented calcified root canal or root resorption. A total of 306 CBCTs remained, of which 208 corresponded to males (67.97%) and 98 corresponded to females (32.03%). Mean age was 44.95 ± 15.74 years, ranging from 18 to 85. The age groups were as follows: 72 subjects (23.53%) were under 30 years of age, 61 (19.93%) between 31 and 40 years, 53 (17.32%) between 41 and 50 years, 65 (21.24%) between 51 and 60 years, and 55 (17.97%) were older than 60 years (Table 1). Intraobserver variability ranged between 0.478 and 0.887, and interobserver variability ranged between 0.403 and 0.865.

### 3.1. Sagittal Linear Measurements and Ratios

All sagittal horizontal measurements (sce_TW, sce_PW, and sph_CW), and chamber length (s_PCL) showed statistically significant differences between age groups (*p* ≤ 0.050). sce_TW, sce_PW, and s_PCL decreased while sph_CW increased with age (Table 1).

All ratios showed statistically significant differences between age groups (*p* ≤ 0.050). s_IL/s_TTL and s_RL/s_TTL increased, while s_PCL/s_TTL, s_PCL plus s_IL/s_TTL and sce_PW/sce_TW decreased with age (Table 1).

### 3.2. Coronal Linear Measurements and Ratios

Mesiodistal pulp length at cementoenamel junction level (cce_MDPL) and all the mesiodistal measurements at pulp horn level (cph_MDTL and cph_MDPL) showed statistically significant differences between age groups (*p* ≤ 0.050). All these measurements decreased with age (Table 2).

The two mesiodistal coronal ratios (cce_MDPL/cce_MDTL and cph_MDPL/cph_MDTL) showed statistically significant differences between age groups (*p* ≤ 0.050). These measurements also decreased with age (Table 2).

### 3.3. Axial Linear Measurements and Ratios

Three axial mesiodistal measurements (ace_MDPL, aph_MDTL, and aph_MDPL) and all palatovestibular measurements (ace_PVTL, ace_PVPL, and aph_PVTL) showed statistically significant differences between age groups (*p* ≤ 0.050). These measurements decreased with age (Table 3).

All ratios showed statistically significant differences between the different age groups (*p* ≤ 0.050). ace_MDPL/ace_MDTL, ace_PVPL/ace_PVTL, ace_PVPL/ace_MDTL, and aph_MDPL/aph_MDTL decreased, while aph_PVTL/aph_MDTL increased with age (Table 3).

### 3.4. Age Estimation Based on Multiplanar Measurements and Ratios

Multiple linear regressions based on sagittal, coronal and axial linear measurements and ratios were independently performed. The equations obtained from sagittal linear measurements and ratios presented a standard error of the estimate (SEE) of ±12.4 years (R^2^ = 0.36), and a SEE of ±12.4 years (R^2^ = 0.37), respectively. The equations obtained from coronal linear measurements and ratios presented a SEE of ±12.3 years (R^2^ = 0.37), and a SEE of ±13.7 years (R^2^ = 0.22), respectively. The equations obtained from axial linear measurements and ratios presented a SEE of ±10.9 years (R^2^ = 0.49), and a SEE of ±10.8 years (R^2^ = 0.50), respectively (Table 4).

Multiple linear regressions considering all the linear measurements and ratios, excluding homonym variables in different slices, were independently performed. The equation obtained from linear measurements presented a SEE of ±10.9 years (R^2^ = 0.52), and the equation from ratios presented a SEE of ±10.7 years (R^2^ = 0.51), (Table 4).

## 4. Discussion

The present study has identified CBCT linear variables which are clinically and statistically acceptable for making age estimation equations for forensic purposes. These equations were derived after 3D analysis of upper central incisors. The most predictable equations were mainly built on axial measurements and ratios. Our findings reveal that horizontal dimensions, mainly pulp dimensions, allow for more accurate age estimation.

Various methods have been employed to indirectly quantify secondary dentin deposition. Early methods using 2D imaging techniques include the Kvaal et al. method [2], the Ikeda et al. tooth coronal index [17], and the Cameriere et al. pulp to tooth area ratio [15]. With respect to 3D radiology, micro-CT is the gold standard in terms of measurement accuracy because it provides higher spatial resolution and, thus, precise segmentation of dental structures [29]. However, micro-CT presents certain drawbacks. It can only be applied on a single tooth in postmortem specimens and image acquisition is time consuming [18]. Conversely, CBCT can be used in multiple teeth in vivo, is less expensive and minimizes radiation exposure [33]. Therefore, CBCT is widely used in dentistry, enabling 3D information on living individuals to be easily acquired.

Starting with Yang et al. [28], various authors have used CBCT for age estimation and reported a wide range of accuracy [19,20,21,24,28,34,35,36,37,38]. While some authors report a low capacity for age estimation [11,28,34,36,39,40], others report equations with good R^2^ and SEE values [19,20,37,38], showing that CBCT can be as useful as CT [26,41,42,43,44], or micro-CT [18,22,31,32]. Various methods involving ratios of dental linear measurements have been employed on CBCT for age estimation, e.g., Kwaal’s method [45,46], Cameriere’s pulp tooth area ratio [19,20,37,43], pulp to tooth volume ratio [31,47,48,49,50], pulp volume [11,38], and chamber volume [29,30,51,52]. The equations obtained in the present study using linear measurements and ratios yield acceptable R^2^ and SEE values. Regression equations in the literature yield R^2^ values ranging from 0.15 to 0.91 [11,19,20,21,24,28,29,30,37,38,39,42] and SEE values ranging from 4.2 to 11.45 years [20,28,29,30,35,37,38]. Apart from differences in measurement method, variability is also due to differences in tooth type. Good age estimation has been reported using CBCT for different teeth such as maxillary canines [19,20,38,42,44], mandibular canines [42,44], maxillary lateral incisors [49], maxillary second premolars [45], maxillary and mandibular first molars [29,53], and maxillary and mandibular second molars [51]. However, most research has been based on uniradicular teeth [11,19,20,21,23,24,25,28,35,37,38], specifically upper central incisors [35,36,37,38,54,55,56] and canines [11,19,20,21,25,34,41,42,43,47]. We have chosen to use the upper central incisor because its pulp has a simple anatomy, considerable size, and few anatomic variations, making it easier to measure than posterior teeth [2]. Secondary dentinal deposition is harder to analyze in teeth with smaller pulp dimensions such asmandibular incisors [15,23,24,25,27]. Another reason for analyzing upper central incisors was that they are one of the most frequent remanent teeth in older patients [2]. Most CBCT research comparing tooth groups found the highest correlation with age for maxillary central incisors [23,24,37,38,46,48,54,57]. As compared to volume methods, the present research based on incisor linear measurements yielded an R^2^ within the previously observed range [35,56], and a slightly higher SEE [30,37]. However, our method is easier and less time consuming than certain volumetric methods (pulp/tooth volume and pulp/crown volume) that may take from 30 min to several hours [55]. Furthermore, our equations are less complex than those involving several teeth and measurements, while yielding similar R^2^ and SEE values [46].

Our results are in line with previous research that successfully used root canal diameter as an indirect quantification method for secondary dentin deposition [31]. A number of authors have reported that dentinal thickness increased with age more along the pulp walls than along the pulp roof [58,59], and our findings concur. Some authors have found that pulp cavity height significantly correlated with chronological age on 2D imaging [2,60], however, the present study found horizontal measurements to have better R^2^ and SEE values than vertical measurements. Our results were consistent with the vast majority of CBCT research [37,45,46,61], except for Lee et al. [20]. Thus, we determined that horizontal dimensions are more accurate for age estimation.

The fact that vertical measurements are scarcely useful for age estimation may be due to the confounding effect of external factors such as attrition, occlusion type, or behavioral habits. Our results support the idea that attrition is not sufficiently related to age to be used for age estimation, which is in line with other authors [2]. Previous reports have already described that attrition bears stronger relation to diet and habits [62], and, thus, the low harshness of the current diet may explain the lack of association with age.

Apart from secondary dentin deposition, pulp cavity dimensions may also be modified by other local and systemic factors, such as tertiary dentin deposition at the chamber roof due to decay or to restorations [6]. To avoid this problem, we have used only sound teeth following Kvaal’s instructions [2].

Some authors have argued that pulp dimensions may be influenced by other factors such as race, nutritional and hormonal changes and certain systemic diseases [63]. The inclusion of patients without analyzing their clinical histories could be considered a shortcoming of the present study. However, this is not the case because our aim was not to analyze the influence of systemic diseases on secondary dentin deposition but to find an equation for age estimation in the general population. Furthermore, secondary dentin deposition is considered to vary by tooth type, sex and population [2,59]. This is one of the reasons why regression-specific equations are necessary to validate age estimation methods in different populations [64,65,66,67]. Although Cameriere et al. [68] were able to adequately determine age for both Italian and Portuguese samples with the same regression equation, specific equations may well yield different age estimates for different populations [25,65]. Thus, future research should consider evaluating our method in other populations.

Forensic dentistry is constantly searching for the ideal age estimation method. Each method presents strengths and weaknesses. Therefore, in order to maximize age estimation accuracy, it may be advisable to apply more than one method and to repeat measurements [2,23,30,42,69]. It may also be advisable to include multiple tooth types [69], which may be seen as a limitation of the present study focusing only on upper central incisors. Nevertheless, there is disagreement among authors regarding multicollinearity problems when using multiple tooth types in a single model [11,69].

## 5. Conclusions

The present study shows that CBCT linear measurements on upper central incisors are an acceptable method for age estimation. Horizontal measurements, mainly pulp measurements, are preferable for improving accuracy. Although CBCT has become an important tool for forensic dentistry, future research should determine whether combining several CBCT age estimation methods could improve results. Automatic software development is likely to make integrating measurement methods faster and more efficient.

## Figures and Tables

**Figure 1 ijerph-19-13370-f001:**
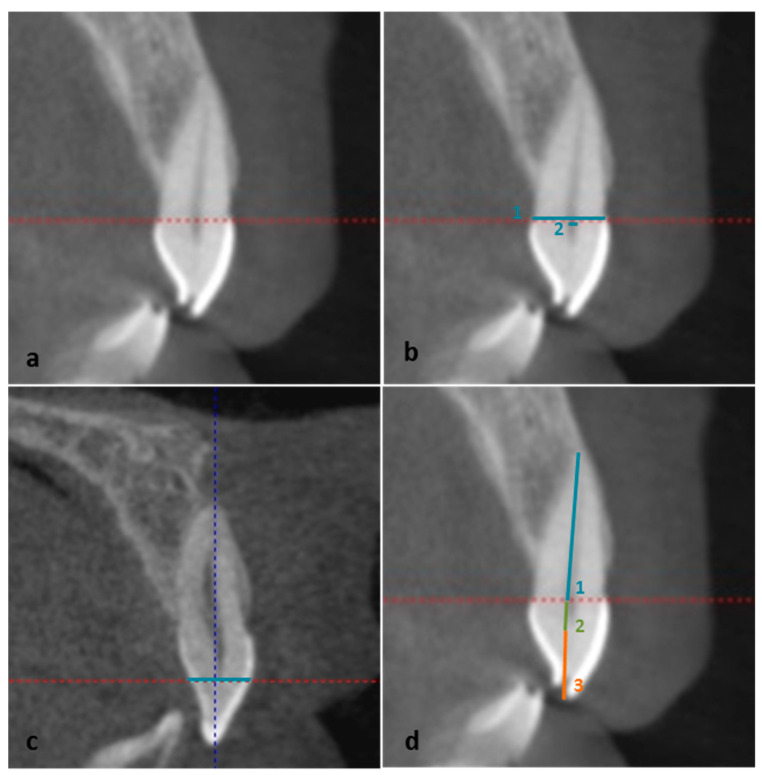
(**a**) Incisor in neutral position; (**b**) sagittal measurements at cementoenamel junction: (1) sagittal tooth width (sce_TW), (2) pulp width (sce_PW); (**c**) sagittal measurement at pulp horn: incisor crown width (sph_CW); and (**d**) sagittal vertical measurements: (1) root length: (s_RL), (2) pulp chamber length (s_PCL) and (3) incisal length: (s_IL).

**Figure 2 ijerph-19-13370-f002:**
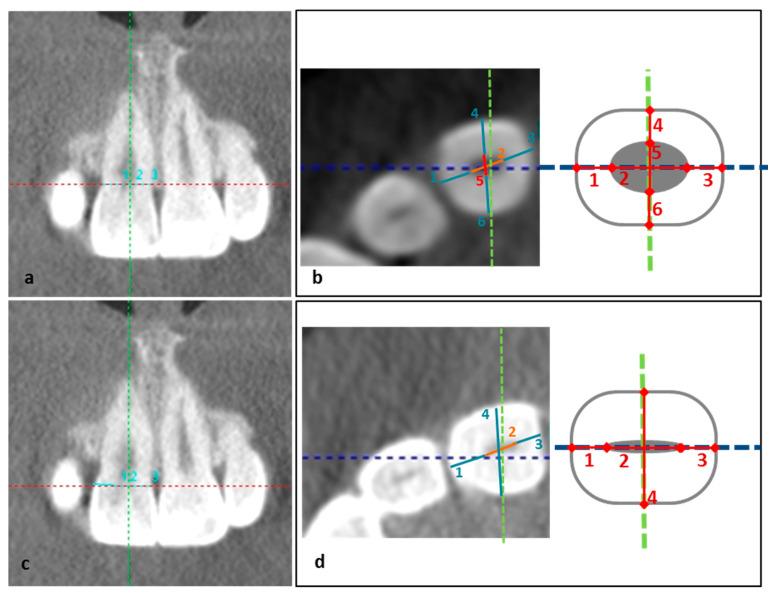
(**a**) Coronal measurements at cementoenamel junction level: mesiodistal tooth length (cce_MDTL, sum of the values 1+2+3), mesiodistal pulp length (cce_MDPL, value 2); (**b**) axial measurements at cementoenamel junction level: mesiodistal tooth length (ace_MDTL, sum of the values 1+2+3); (**c**) coronal measurements at pulp horn level: mesiodistal pulp length (cph_MDPL, value 2), palatovestibular tooth length (ace_PVTL, sum of values 4+5+6), palatovestibular pulp length (ace_PVPL, value 5) and (**d**) axial measurements at pulp horn level: mesiodistal tooth length (aph_MDTL,sum of the values 1+2+3), mesiodistal pulp length (aph_MDPL, value 2), palatovestibular tooth length (aph_PVTL, value 4).

**Table 1 ijerph-19-13370-t001:** Differences in sagittal linear measurements and ratios between age groups.

	Age Groups	Mean ^1^	Min.	Max.	F	*p* *
sce_TW	≤30	7.26 ± 0.61 ^a^	5.52	8.80	4.479	0.002 *
	31–40	7.24 ± 0.50 ^b^	6.00	8.25		
	41–50	7.27 ± 0.44 ^c^	6.00	8.25		
	51–60	7.11 ± 0.50	5.75	8.25		
	>60	6.93 ± 0.48 ^a,b,c^	6.00	8.70		
	Total	7.17 ± 0.53	5.52	8.80		
sce_PW	≤30	1.78 ± 0.38 ^a^	1.00	3.00	19.613	<0.001 *
	31–40	1.62 ± 0.38 ^b^	1.00	3.30		
	41–50	1.67 ± 0.37 ^c^	1.00	2.75		
	51–60	1.43 ± 0.30 ^a,b,c^	0.50	2.00		
	>60	1.28 ± 0.30 ^a,b,c^	0.75	2.00		
	Total	1.57 ± 0.39	0.50	3.30		
sph_CW	≤30	6.09 ± 0.94 ^a^	4.50	8.75	12.335	<0.001 *
	31–40	6.49 ± 0.76 ^b^	5.10	8.50		
	41–50	6.58 ± 0.83 ^a^	4.75	8.10		
	51–60	6.92 ± 0.72 ^a,b^	5.50	8.75		
	>60	6.94 ± 0.78 ^a,b^	5.00	8.70		
	Total	6.58 ± 0.87	4.50	8.75		
s_RL	≤30	12.71 ± 1.75	8.54	16.64	1.607	0.172
	31–40	13.16 ± 2.09	6.00	17.44		
	41–50	13.22 ± 2.03	7.84	17.25		
	51–60	13.39 ± 1.33	10.03	16.61		
	>60	13.43 ± 2.10	7.51	18.06		
	Total	13.16 ± 1.87	6.00	18.06		
s_PCL	≤30	3.87 ± 0.95 ^a^	1.82	5.75	32.099	<0.001 *
	31–40	3.57 ± 0.85 ^b^	1.27	5.41		
	41–50	3.25 ± 1.15 ^a,c^	0.60	6.02		
	51–60	2.57 ± 1.02 ^a,b,c^	0.90	6.50		
	>60	2.09 ± 1.06 ^a,b,c^	0.25	5.75		
	Total	3.12 ± 1.19	0.25	6.50		
s_IL	≤30	6.26 ± 1.00	3.06	8.49	2.043	0.088
	31–40	6.55 ± 1.16	0.93	8.92		
	41–50	6.79 ± 1.37	3.75	10.61		
	51–60	6.78 ± 1.36	3.51	10.00		
	>60	6.75 ± 1.59	1.25	10.01		
	Total	6.61 ± 1.30	0.93	10.61		
s_TTL	≤30	22.90 ± 2.23	17.52	27.80	2.206	0.068
	31–40	23.28 ± 2.23	15.08	28.21		
	41–50	23.26 ± 2.37	15.84	27.71		
	51–60	22.70 ± 2.14	18.14	31.34		
	>60	22.20 ± 2.32	16.76	27.20		
	Total	22.87 ± 2.27	15.08	31.34		
sce_PW/sce_TW	<30	0.25 ± 0.05 ^a^	0,14	0.41	16.439	<0.001 *
	31–40	0.22 ± 0.05 ^b^	0.14	0.44		
	41–50	0.23 ± 0.05 ^c^	0.14	0.33		
	51–60	0.20 ± 0.04 ^a,c^	0.07	0.30		
	>60	0.19 ± 0.04 ^a,b,c^	0.10	0.27		
	Total	0.22 ± 0.05	0.07	0.44		
s_IL/s_TTL	<30	0.27 ± 0.04 ^a^	0.15	0.38	2.932	0.021 *
	31–40	0.28 ± 0.05	0.04	0.41		
	41–50	0.29 ± 0.06	0.18	0.46		
	51–60	0.30 ± 0.05	0.16	0.40		
	>60	0.31 ± 0.08 ^a^	0.06	0.48		
	Total	0.29 ± 0.06	0.04	0.48		
s_PCL/s_TTL	<30	0.17 ± 0.04 ^a^	0.08	0.25	29.477	<0.001 *
	31–40	0.15 ± 0.04 ^b^	0.05	0.24		
	41–50	0.14 ± 0.05 ^a,c^	0.03	0.26		
	51–60	0.11 ± 0.04 ^a,b,c^	0.04	0.21		
	>60	0.10 ± 0.05 ^a,b,c^	0.01	0.29		
	Total	0.14 ± 0.05	0.01	0.29		
s_RL/s_TTL	<30	0.56 ± 0.05 ^a^	0.42	0.69	11.183	<0.001 *
	31–40	0.56 ± 0.05 ^b^	0.40	0.79		
	41–50	0.57 ± 0.05 ^c^	0.47	0.68		
	51–60	0.59 ± 0.04 ^a,b^	0.49	0.72		
	>60	0.60 ± 0.05 ^a,b,c^	0.45	0.75		
	Total	0.57 ± 0.05	0.40	0.79		
(s_PCL plus s_IL)/s_TTL	<30	4.15 ± 0.94 ^a^	2.15	5.96	33.172	<0.001 *
	31–40	3.85 ± 0.83 ^b^	1.62	5.63		
	41–50	3.54 ± 1.10 ^a,c^	1.06	6.20		
	51–60	2.87 ± 0.98 ^a,b,c^	1.29	6.81		
	>60	2.40 ± 1.01 ^a,b,c^	0.57	5.81		
	Total	3.40 ± 1.16	0.57	6.81		

^1^ Values expressed as mean ± SD; * ANOVA test; ^a,b,c^ Bonferroni post hoc test (paired letters showed statistical significance). Sce_TW: tooth width at cementoenamel junction level, sce_PW: pulp width at cementoenamel junction level, sph_CW: incisor crown at pulp horn level, s_RL: root length, s_PCL: pulp chamber length, s_IL: incisal length, s_TTL: total tooth length.

**Table 2 ijerph-19-13370-t002:** Differences in coronal linear measurements and ratios between age groups.

	Age Groups	Mean ^1^	Min.	Max.	F	*p* *
cce_MDTL	≤30	6.37 ± 0.68	4.75	7.80	1.218	0.303
	31–40	6.36 ± 0.61	4.50	7.50		
	41–50	6.47 ± 0.49	5.50	7.50		
	51–60	6.40 ± 0.56	5.10	7.80		
	>60	6.23 ± 0.51	5.00	7.20		
	Total	6.36 ± 0.58	4.50	7.80		
cce_MDPL	≤30	1.97 ± 0.36 ^a^	1.50	2.80	15.110	<0.001 *
	31–40	1.98 ± 0.44 ^b^	1.20	3.00		
	41–50	1.95 ± 0.39 ^c^	1.00	3.00		
	51–60	1.75 ± 0.36 ^a,b,c,d^	1.00	2.80		
	>60	1.54 ± 0.30 ^a,b,c,d^	0.75	2.40		
	Total	1.85 ± 0.41	0.75	3.00		
cph_MDTL	≤30	8.06 ± 0.72 ^a^	6.00	9.50	21.773	<0.001 *
	31–40	7.81 ± 0.73 ^b^	6.25	9.60		
	41–50	7.74 ± 0.95 ^c^	4.50	10.25		
	51–60	7.26 ± 0.77 ^a,b,c^	6.00	9.00		
	>60	6.89 ± 0.78 ^a,b,c^	5.25	8.75		
	Total	7.57 ± 0.89	4.50	10.25		
cph_MDPL	≤30	2.43 ± 0.56 ^a^	1.50	3.60	33.935	<0.001 *
	31–40	2.26 ± 0.52 ^b^	1.50	3.75		
	41–50	2.18 ± 0.44 ^a,c^	1.25	3.50		
	51–60	1.82 ± 0.48 ^a,b,c,d^	0.75	3.25		
	>60	1.49 ± 0.41 ^a,b,c,d^	0.75	2.50		
	Total	2.07 ± 0.59	0.75	3.75		
cce_MDPL/cce_MDTL	<30	0.31 ± 0.04 ^a^	0.23	0.42	16.38	<0.001 *
	31–40	0.31 ± 0.06 ^b^	0.19	0.43		
	41–50	0.30 ± 0.05 ^c^	0.18	0.46		
	51–60	0.27 ± 0.06 ^a,b,d^	0.14	0.44		
	>60	0.25 ± 0.05 ^a,b,c,d^	0.13	0.33		
	Total	0.29 ± 0.06	0.13	0.46		
cph_MDPL/cph_MDTL	<30	0.30 ± 0.06 ^a^	0.17	0.42	18.30	<0.001 *
	31–40	0.29 ± 0.06 ^b,^	0.19	0.43		
	41–50	0.29 ± 0.06 ^c^	0.15	0.44		
	51–60	0.25 ± 0.07 ^a,b,d^	0.11	0.48		
	>60	0.22 ± 0.06 ^a,b,c,d^	0.11	0.33		
	Total	0.27 ± 0.07	0.11	0.48		

^1^ Values expressed as mean ± SD, * ANOVA test; ^a,b,c,d^ Bonferroni post hoc test (paired letters showed statistical significance). Cce_MDTL: mesiodistal tooth length, cce_MDPL: mesiodistal pulp length at cementoenamel junction level, cph_MDTL: mesiodistal tooth length at pulp horn level, cph_MDPL: mesiodistal pulp length at pulp horn level.

**Table 3 ijerph-19-13370-t003:** Differences in axial linear measurements and ratios between age groups.

	Age Groups	Mean ^1^	Min.	Max.	F	*p* *
ace_MDTL	≤30	6.57 ± 0.63	4.82	7.90	1.246	0.292
	31–40	6.55 ± 0.66	4.44	8.14		
	41–50	6.72 ± 0.62	5.39	8.25		
	51–60	6.59 ± 0.54	5.44	8.12		
	>60	6.47 ± 0.58	5.19	7.95		
	Total	6.58 ± 0.61	4.44	8.25		
ace_MDPL	≤30	2.21 ± 0.45 ^a^	1.25	3.22	10.695	<0.001 *
	31–40	2.23 ± 0.49 ^b^	1.25	3.35		
	41–50	2.23 ± 0.51 ^c^	1.00	3.25		
	51–60	2.02 ± 0.51 ^d^	0.75	3.30		
	>60	1.75 ± 0.44 ^a,b,c,d^	0.79	2.85		
	Total	2.10 ± 0.51	0,75	3.35		
ace_PVTL	≤30	7.59 ± 0.60 ^a^	6.25	9.20	3.180	0.014 *
	31–40	7.50 ± 0.46	6.60	8.71		
	41–50	7.58 ± 0.53	6.50	9.51		
	51–60	7.39 ± 0.52	6.25	8.51		
	>60	7.30 ± 0.54 ^a^	6.25	8.70		
	Total	7.48 ± 0.54	6.25	9.51		
ace_PVPL	≤30	1.90 ± 0.36 ^a^	1.00	3.25	16.819	<0.001 *
	31–40	1.78 ± 0.32 ^b^	1.00	2.72		
	41–50	1.71 ± 0.33 ^c^	1.25	2.50		
	51–60	1.61 ± 0.29 ^a,b^	1.00	2.40		
	>60	1.44 ± 0.34 ^a,b,c^	0.75	2.25		
	Total	1.70 ± 0.36	0.75	3.25		
aph_MDTL	≤30	8.24 ± 0.64 ^a^	6.72	9.70	23.602	<0.001 *
	31–40	7.94 ± 0.72 ^b^	6.18	9.19		
	41–50	7.85 ± 0.86 ^a,c^	6.02	10.36		
	51–60	7.43 ± 0.71 ^a,b,c^	5.95	8.83		
	>60	7.09 ± 0.74 ^a,b,c^	5.70	8.48		
	Total	7.73 ± 0.83	5.70	10.36		
aph_MDPL	≤30	3.02 ± 0.56 ^a^	2.00	4.32	50.491	<0.001 *
	31–40	2.70 ± 0.67 ^a,b^	1.20	3.91		
	41–50	2.28 ± 0.61 ^a,b,c^	0.75	3.40		
	51–60	1.99 ± 0.67 ^a,b,d^	0.50	3.40		
	>60	1.61 ± 0.49 ^a,b,c,d^	0.50	2.75		
	Total	2.37 ± 0.79	0.50	4.32		
aph_PVTL	≤30	5.85 ± 1.03 ^a^	4.00	9.20	14.437	<0.001 *
	31–40	6.25 ± 0.80 ^b^	4.50	7.81		
	41–50	6.39 ± 1.01 ^a,c^	3.00	8.00		
	51–60	6.78 ± 0.75 ^a,b,^	5.25	9.01		
	>60	6.88 ± 0.78 ^a,b,c^	4.76	8.50		
	Total	6.41 ± 0.96	3.00	9.20		
ace_MDPL/ace_MDTL	<30	0.34 ± 0.05 ^a^	0.19	0.42	12.269	<0.001 *
	31–40	0.34 ± 0.06 ^b^	0.25	0.49		
	41–50	0.33 ± 0.06 ^c^	0.15	0.44		
	51–60	0.31 ± 0.07 ^b,d^	0.10	0.44		
	>60	0.27 ± 0.06 ^a,b,c,d^	0.12	0.43		
	Total	0.32 ± 0.07	0.10	0.49		
ace_PVPL/ace_PVTL	<30	0.25 ± 0.05 ^a^	0.15	0.48	13.323	<0.001 *
	31–40	0.24 ± 0.04 ^b^	0.14	0.36		
	41–50	0.23 ± 0.04 ^a^	0.16	0.33		
	51–60	0.22 ± 0.04 ^a^	0.13	0.30		
	>60	0.20 ± 0.04 ^a,b,c^	0.10	0.29		
	Total	0.23 ± 0.05	0.10	0.48		
ace_PVPL/ace_MDTL	<30	0.29 ± 0.07 ^a^	0.14	0.60	12.464	<0.001 *
	31–40	0.27 ± 0.05 ^b^	0.13	0.42		
	41–50	0.26 ± 0.05 ^a,c^	0.16	0.39		
	51–60	0.25 ± 0.05 ^a^	0.13	0.38		
	>60	0.23 ± 0.06 ^a,b,c^	0.13	0.37		
	Total	0.26 ± 0.06	0.13	0.60		
aph_MDPL/aph_MDTL	<30	0.37 ± 0.06 ^a^	0.23	0.48	37.342	<0.001 *
	31–40	0.34 ± 0.07 ^b^	0.16	0.48		
	41–50	0.29 ± 0.07 ^a,b,c^	0.09	0.41		
	51–60	0.27 ± 0.08 ^a,b,d^	0.07	0.42		
	>60	0.23 ± 0.06 ^a,b,c,d^	0.08	0.36		
	Total	0.30 ± 0.09	0.07	0.48		
aph_PVTL/aph_MDTL	<30	0.71 ± 0.13 ^a^	0.48	1.05	33.246	<0.001 *
	31–40	0.80 ± 0.15 ^a,b^	0.52	1.25		
	41–50	0.83 ± 0.16 ^a,c^	0.30	1.20		
	51–60	0.92 ± 0.14 ^a,b,c^	0.59	1.26		
	>60	0.98 ± 0.15 ^a,b,c^	0.60	1.26		
	Total	0.84 ± 0.17	0.30	1.26		

^1^ Values expressed as mean ± SD, * ANOVA test; ^a,b,c,d^ Bonferroni post hoc test (paired letters showed statistical significance). Ace_MDTL: mesiodistal tooth length at cementoenamel junction level, ace_MDPL mesiodistal pulp length at cementoenamel junction level, ace_PVTL: palatovestibular tooth length at cementoenamel junction level, ace_PVPL palatovestibular pulp length at cementoenamel junction level, aph_MDTL mesiodistal tooth length at pulp horn level, aph_MDPL: mesiodistal pulp length at pulp horn level, aph_PVTL: palatovestibular tooth length at pulp horn level.

**Table 4 ijerph-19-13370-t004:** Multiple linear regression models for age estimation based on upper central incisors.

	Adjusted R^2^	SEE ^1^ (±year)	*p* *
**SAGITTAL**			
Linear measurements			
Age=68.050−5.828 s_PCL−11.509 sce_PW+0.987 s_RL	0.364	12.495	0.011 *
Ratios			
$\mathrm{Age}=55.463-5.685\frac{s\_\mathrm{PCL}\;\mathrm{plus}\;s\_\mathrm{IL}}{s\_\mathrm{TTL}}-80.909\frac{\mathrm{sce}\_\mathrm{PW}}{\mathrm{sce}\_\mathrm{TW}}+46.024\frac{s\_\mathrm{RL}}{s\_\mathrm{TTL}}$	0.372	12.417	0.003 *
**CORONAL**			
Linear measurements			
Age=75.844−8.630 cph_MDPL−5.863 cph_MDTL +6.953 cce_MDTL−7.250 cce_MDPL	0.373	12.320	0.001 *
Ratios			
$\mathrm{Age}=86.030-72.205\frac{\mathrm{cph}\_\mathrm{MDPL}}{\mathrm{cph}\_\mathrm{MDTL}}-75.696\frac{\mathrm{cce}\_\mathrm{MDPL}}{\mathrm{cce}\_\mathrm{MDTL}}$	0.225	13.701	0.000 *
**AXIAL**			
Linear measurements			
Age=72.963−10.084 aph_MDPL−12.017 ace_PVPL+2.477 aph_PVTL	0.498	10.957	0.001 *
Ratios			
$\mathrm{Age}=68.762-68.668\frac{\mathrm{aph}\_\mathrm{MDPL}}{\mathrm{aph}\_\mathrm{MDTL}}+26.912\frac{\mathrm{aph}\_\mathrm{PVTL}}{\mathrm{aph}\_\mathrm{MDTL}}-61.559\frac{\mathrm{ace}\_\mathrm{PVPL}\;}{\mathrm{ace}\_\mathrm{MDTL}}-31.265\frac{\mathrm{ace}\_\mathrm{MDPL}\;}{\mathrm{ace}\_\mathrm{MDTL}}$	0.509	10.839	0.003 *
**MULTIPLANAR MEASUREMENTS** (homonyms excluded)			
Linear measurements			
Age=80.608−9.197 aph_MDPL−10.310 ace_PVPL−2.765 s_PCL+0.901 s_RL	0.520	10.717	0.000 *
Ratios			
$\mathrm{Age}=107.129-69.226\frac{\mathrm{aph}\_\mathrm{MDPL}}{\mathrm{aph}\_\mathrm{MDTL}}-4.195\frac{s\_\mathrm{PCL}\;\mathrm{plus}\;s\_\mathrm{IL}}{s\_\mathrm{TTL}}-73.910\frac{\mathrm{ace}\_\mathrm{PVPL}\_p}{\mathrm{ace}\_\mathrm{PVTL}}-32.828\frac{\mathrm{ace}\_\mathrm{MDPL}}{\mathrm{ace}\_\mathrm{MDTL}}$	0.517	10.757	0.000 *

^1^ Standard error of estimate, * Multivariant linear regression analysis. Sce_TW: tooth width at cementoenamel junction level, sce_PW: pulp width at cementoenamel junction level, s_RL: root length, s_PCL: pulp chamber length, s_TTL: total tooth length, cce_MDTL: mesiodistal tooth length, cce_MDPL: mesiodistal pulp length at cementoenamel junction level, cph_MDTL: mesiodistal tooth length at pulp horn level, cph_MDPL: mesiodistal pulp length at pulp horn level, ace_MDTL: mesiodistal tooth length at cementoenamel junction level, ace_MDPL mesiodistal pulp length at cementoenamel junction level, ace_PVTL: palatovestibular tooth length at cementoenamel junction level, ace_PVPL palatovestibular pulp length at cementoenamel junction level, aph_MDTL mesiodistal tooth length at pulp horn level, aph_MDPL: mesiodistal pulp length at pulp horn level, aph_PVTL: palatovestibular tooth length at pulp horn level.

## Data Availability

The data presented in this study are available on request from the corresponding author.

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
