# Peer review of "Age Estimation Using Maxillary Central Incisor Analysis on Cone Beam Computed Tomography Human Images"

_ijerph, 2022, doi:10.3390/ijerph192013370_

Round 1

Reviewer 1 Report

Dear Authors,

Your article entitled Age estimation using maxillary central incisor analysis on Cone Beam Computed Tomography human images brings useful information for the field of forensic medicine.

I have two recommendations to make for the further improvement of the paper:

-please mention the study type in the Materials and methods section

-Figure 2, line 146 - please consider replacing "b)" with "d)".

Author Response

Dear Ms. Addie GE

Thanks for your feedback. We are truly grateful for yours and reviewer’s comments. We are convinced that these modifications will improve the quality of the paper.

Following the instructions of both reviewers, we have made the following modifications: Comments to the Author

-Reviewer  #1

  1. English language and style: English language and style are fine/minor spell check required.

Response: We have succintly checked English language to improve manuscript style. Now, we have included serveral minor changes along the text.

  1. I have two recommendations to make for the further improvement of the paper:
  2. a) please mention the study type in the Materials and methods section

Response: Following the suggestion made by reviewer #1, we have modified the following paragraph including mention of the study type. The paragraph “A total of 373 CBCTs were randomly selected from the Radiology Unit of the Medicine and Dentistry School at the Santiago de Compostela University. These CBCTs were performed from 2012 to 2017 for treatment planning of various oral surgical procedures” has modified as follows: “A total of 373 randomly selected CBCTs were included in this cross-sectional study. Sample size was determined by means of the Slovin’s formula. These CBCTs were performed from 2012 to 2017 for treatment planning of various oral surgical procedures in the Radiology Unit of the Medicine and Dentistry School at the Santiago de Compostela University”.

  1. b) Figure 2, line 146 - please consider replacing "b)" with "d)".

Response: We have considered the suggestion by reviewer #1. We have modified Figure 2, line 146. We have replaced “b)” with “d)”. The paragraph “Figure 2. (a) Coronal measurements at cementoenamel junction level, (b) axial measurements at cementoenamel junction level, (c) Coronal measurements at pulp horn level, and (b) axial measurements at pulp horn level” has been modified by the following paragraph: Figure 2. (a) Coronal measurements at cementoenamel junction level, (b) axial measurements at cementoenamel junction level, (c) Coronal measurements at pulp horn level, and (d) axial measurements at pulp horn level”.

Reviewer 2 Report

After critically reviewing this Research Article titled "Age estimation using maxillary central incisor analysis on Cone Beam Computed Tomography human images", I detected some points that need to revise, after revision the manuscript can be further considered if is it suitable for acceptance for publication. Below please find my detailed comments.

Title and Abstract

-Please unify the letter case (e.g., A or a) of the English letters of the title

-The information of the corresponding author should include mail address and tel.

The introduction includes a brief and clear background of this field but it is suggested to add the hypothesis of this study at the end of the Introduction.

Materials and Method

- The test for normality distribution and homogeneity is required.

- How did the authors determine the number of specimens? 

Result

- Line 173 and 181, “table 1” should be “Table 1,” and please confirm the writing format of the entire article.

- Table 1, “*Anova’s test”, this is not correct, please revise.

- Table 2, I am curious is there no statistical difference between all the groups in s_RL, s_IL, and s_TTL? please verify again. and also some groups in Table 3.

- Table 4, why is there no marks of P values (such as * mark)?

The discussion is voluminous enough and the conclusion is concise and shows the main conclusions of the study. 

References: there is a very sufficient number of references, but some of them are very old, if it is necessary, the author can keep them, but if it is not that important, please delete some outdated references.

Author Response

Dear Ms. Addie GE

Thanks for your feedback. We are truly grateful for yours and reviewer’s comments. We are convinced that these modifications will improve the quality of the paper.

Following the instructions of both reviewers, we have made the following modifications: Comments to the Author

-Reviewer  #2

After critically reviewing this Research Article titled "Age estimation using maxillary central incisor analysis on Cone Beam Computed Tomography human images", I detected some points that need to revise, after revision the manuscript can be further considered if is it suitable for acceptance for publication. Below please find my detailed comments.

  1. English language and style: English language and style are fine/minor spell check required.

Response: We have rechecked English language to improve manuscript style. Now, we have included serveral minor changes along the text. 

  1. Title and Abstract

-Please unify the letter case (e.g., A or a) of the English letters of the title

Response: Following the recommendations of reviewer #2, we have unified the letter case of the English letter of the title. Now the title is the following: “Age estimation using maxillary central incisor analysis on cone beam computed tomography human images”.

-The information of the corresponding author should include mail address and tel.

Response. Now, we have included the phone number of the corresponding author following her email address in line 16. Now, this line is as follows: Correspondence: mariamercedes.suarez@usc.es, Tel.: +34881812437

CorrespondingAuthor:

María Mercedes Suárez-Cunqueiro

Associate Professor

Research Institute of Santiago de Compostela (IDIS)

Department of Surgery and Medical-Surgical Specialties

Medicine and Dentistry School

C/ Entrerrios S/N 15872

Universidade de Santiago de Compostela, Spain

Tel: +34881812437

e-mail: mariamercedes.suarez@usc.es

  1. The introduction includes a brief and clear background of this field but it is suggested to add the hypothesis of this study at the end of the Introduction.

Response:  Thank you very much for your comment. Now, we have added the following sentence at the end of the Introduction in line 66 to explain the hypothesis of this study: “We hypothesized that CBCT analysis of secondary dentin deposition by means of multiplanar linear measurements could yield convenient and useful method for age estimation.”.

  1. Materials and Method

- The test for normality distribution and homogeneity is required.

Response: We agree with reviewer #2 observation. We have performed test for normality and homogeneity distributions at the beginning of statistical analysis to select parametric or non-parametric test for analyzing the data. In the initial study design we have already checked normality of the study sample using Kolmogorov-Smirnov test and homogeneity distribution using Levene test. Normality and homogeneity were assumed with a p > 0.05. Now, we have added the following sentence in Materials and Method in line 158: “Kolmogorov-Smirnov test and Levene test were applied to check for normality and homogeneity distribution of the sample”.

- How did the authors determine the number of specimens? 

Response: We have employed the Slovin’s formula to estimate the number of specimens. This equation is applied when size of population is known but has no idea how the population characteristics behave.

  • Sample size = N / (1 + N*e2) = 3500 / (1 + 3500*0,052) = 358.97
    • N= size of population analyzed (We have taken into consideration the 3500 CBCTs available from Radiology Unit of the Medicine and Dentistry School at the Santiago de Compostela University)
    • e= margin of error of 5% (0.05)

Now, following reviewer indications we have included the following information in the Material and Methods section: “Sample size was determined employing the Slovin’s formula.”

  1. Results

- Line 173 and 181, “table 1” should be “Table 1,” and please confirm the writing format of the entire article.

Response: Following the instructions of the Reviewer #2 for lines 173 and 181, we have modified “table 1” for “Table 1”. Also, we have revised the writing format of the full article. We have modified “table 2” for “Table 2” in lines 187 and 190, and “table 3” for “Table 3” in line 200.

- Table 1, “*Anova’s test”, this is not correct, please revise.

Response: Following the instructions, we have modified “*Anova’s test” for “*ANOVA test”. Also, we have done the same for Table 2 and Table 3.

- Table 2, I am curious is there no statistical difference between all the groups in s_RL, s_IL, and s_TTL? please verify again. and also some groups in Table 3.

Response: We want to clarify that SAGITTAL measures are presented in table 1, not in table 2. We appreciate the observation of Reviewer #2. Various authors have found similar results regarding some of these measurements. No statistical differences regarding age were reported in the following manuscripts:

  • Samir E. Bishara; Leigh Vonwald; Jane R. Jakobsen Changes in root length from early to mid-adulthood: Resorption or apposition? 1999, 115(5), 563–568. doi:10.1016/s0889-5406(99)70281-7 
  • Marjorie A. Woods, Quinton C. Robinson, Edward R. Harris. Age progressive changes in pulp widths and root lenghs during adulthood: a study s american blacks and whites. 1990, 9(2), 41-50. doi: 10.1111/j.1741-2358.1990.tb00257.x.

Also, we have verified in Table 3 that the only measurement that not showed a statistical difference between groups was ace_MDTL. This finding was as expected because it is a linear measurement of mesiodistal tooth length at cementoenamel junction in axial slices, and this is a very constant region.

Pearson correlation showed that all the measurements correlated with age with the exception of s_TTL (in line with the manuscripts previously cited), and cce_MDTL and ace_MDTL (homonim measurements in coronal and axial slices). These results support that cce_MDTL and ace_MDTL do not change with increasing age.

- Table 4, why is there no marks of P values (such as * mark)?

Response: We have included the marks of P values in Table 4 as Reviewer #2 suggests.

  1. The discussion is voluminous enough and the conclusion is concise and shows the main conclusions of the study. 

Response: Thank you very much for your comment. We sincerely appreciate that the reviewer #2 was satisfied with the content of the discussion as well as the conclusions.

  1. References: there is a very sufficient number of references, but some of them are very old, if it is necessary, the author can keep them, but if it is not that important, please delete some outdated references.

Response: We are agreeing with your comment. Now, we have deleted the following references from the original manuscript (some of the references have been removed because they are outdated references as Reviewer #2 suggests):

  1. Limdiwala, P.G.; Shah, J.S. Age estimation by using dental radiographs. J. Forensic Dent. Sci. 2013, 5, 118- 122. doi.org/10.4103/0975-1475.119778
  2. Wu, Y.; Niu, Z.; Yan, S.; Zhang, J.; Shi, S.; Wang, T. Age estimation from root diameter and root canal diameter of maxillary central incisors in Chinese Han population using cone-beam computed tomography. Int. J. Clin. Exp. Med. 2016, 9, 9467-9472.
  3. Woods, M.A.; Robinson, Q.C.; Harris, E.F. Age-progressive changes in pulp widths and root lengths during adulthood: a study of American blacks and whites. Gerodontology 1990, 9, 41-50. doi.org/10.1111/j.1741-2358.1990.tb00257.x
  4. Atar, M.; Korperich, E.J. Systemic disorders and their influence on the development of dental hard tissues: a literature review. J. Dent. 2010, 38, 296-306. doi.org/10.1016/j.jdent.2009.12.001
  5. Nitzan, D.W.; Michaeli, Y.; Weinreb, M. Azaz, B. The effect of aging on tooth morphology: a study on impacted teeth. Oral Surg Oral Med Oral Pathol. 1986, 61, 54-60. doi.org/10.1016/0030-4220(86)90203-3.

Round 2

Reviewer 2 Report

The suggested corrections were accepted and the article, in the opinion of this reviewer, is ready for publication.